# Active Cellular and Subcellular Targeting of Nanoparticles for Drug Delivery

**DOI:** 10.3390/pharmaceutics11100543

**Published:** 2019-10-18

**Authors:** Okhil K. Nag, James B. Delehanty

**Affiliations:** Center for Bio/Molecular Science and Engineering, Naval Research Laboratory, Code 6900, 4555 Overlook Ave. SW, Washington, DC 20375, USA

**Keywords:** nanoparticles, active targeted drug delivery, cellular, subcellular, organelles, tumor, cancer, rheumatoid arthritis, Alzheimer’s disease

## Abstract

Nanoparticle (NP)-mediated drug delivery (NMDD) for active targeting of diseases is a primary goal of nanomedicine. NPs have much to offer in overcoming the limitations of traditional drug delivery approaches, including off-target drug toxicity and the need for the administration of repetitive doses. In the last decade, one of the main foci in NMDD has been the realization of NP-mediated drug formulations for active targeted delivery to diseased tissues, with an emphasis on cellular and subcellular targeting. Advances on this front have included the intricate design of targeted NP-drug constructs to navigate through biological barriers, overcome multidrug resistance (MDR), decrease side effects, and improve overall drug efficacy. In this review, we survey advancements in NP-mediated drug targeting over the last five years, highlighting how various NP-drug constructs have been designed to achieve active targeted delivery and improved therapeutic outcomes for critical diseases including cancer, rheumatoid arthritis, and Alzheimer’s disease. We conclude with a survey of the current clinical trial landscape for active targeted NP-drug delivery and how we envision this field will progress in the near future.

## 1. Introduction

One of the persistent challenges in therapeutic drug delivery is the targeted localization of maximal amounts of the therapeutic to the cellular/subcellular site of action in pathologically affected tissues. Before reaching their intended site of action, drug molecules encounter many challenges including a potential chemical breakdown in the intracellular and systemic compartments, crossing various biological barriers (e.g., endothelial cell lining, blood-brain barrier), and remaining pharmacologically active [1,2,3,4]. Historically, these aforementioned issues have been circumvented by administering large quantities of the drug, most often via repetitive dosing regimens which can lead to off-target toxicity, patient noncompliance, and increased healthcare costs [5,6,7]. Thus, a drug that can be preferentially transported to and concentrated at the site of action in disease-affected tissues/cells, while minimizing accumulation in non-targeted normal tissues, is highly desired in modern drug therapy. However, the efficient targeted delivery of drugs remains a considerable challenge on the road to maximizing drug efficacy [3].

Over the past ~20 years, nanoparticles (NPs) have emerged as a promising platform for the improvement of multiple aspects of drug delivery [1,7,8,9]. A search of PubMed for ‘nanoparticle drug delivery’ over this time returns >43,000 citations. Further, as of August 2019, 74 clinical trials including the term “nanoparticle” were listed as “completed” and another 57 were “active” or “recruiting” on ClinicalTrials.gov. There are also many NP-based drug delivery systems in the early stages of preclinical research and development [10]. The major driving force for the use of NPs for improved drug delivery arises from the physicochemical properties of NPs that have much to offer in the improvement of drug efficacy [1,11]. These include: (1) the ability of NPs to carry a significant amount of cargo load (whether appended to the NP surface or loaded in the NP core) [12], (2) NPs’ small size for deep tissue penetration and efficient clearance, (3) the ability to decorate NPs with targeting moieties such as peptides and proteins, and (4) various strategies for the controlled release of NP-associated drug cargos (e.g., light stimulation, pH- or enzyme-triggered release, or the application of an external magnetic field) [13]. Cumulatively, these attributes can help overcome many of the inherent liabilities of current drug formulations including limited or no water-solubility, poor biodistribution, poor pharmacokinetics, higher toxicity, and immunogenic issues [11,14,15].

The delivery of NP-based drug (NP-drug) formulations, however, depends highly on the type of NP, drug, and disease to be treated. Multiple types of surface-modified organic NPs (liposomes, lipids, polymeric, etc.) and inorganic/organic hybrid NPs (metal nanocrystals, carbon dots, etc.) have been described in the literature [16,17,18]. Until now, many NP-drug delivery strategies have relied on the enhanced permeability and retention (EPR) effect [18,19,20]. This passive delivery approach relies on the tendency of materials (including NPs) to accumulate in tumor tissue preferentially over normal tissue due to the ‘leakiness’ of the newly forming vessel architecture in the tumor environment. This type of passive targeting has been used extensively for the delivery of drug carrying-liposomes to cancer tissue, however, only a small percentage (typically <1%) of the total administered NPs reaches the tumor site [21]. Hence the need for robust means of targeting the NP-drug carriers to their needed site of action.

In order to take full advantage of the unique properties offered by NP-drug systems, the ensemble materials must be effectively targeted and delivered to the desired tissues and subsequent cellular/subcellular locations [22,23,24]. Accordingly, a great deal of attention has been given to the fabrication and characterization of novel NP-drug constructs for targeted drug delivery [11]. The design, construction, and implementation of targeted NP-drug complexes is intricate and depends on a number of factors: (1) the type of disease being targeted, (2) the location of the diseased organs/tissues/cells, (3) the strategy for targeting the NP-drug complex to the site, and (4) the mechanism of drug delivery/release. Thus, biological and/or chemical approaches which aim to actively accumulate NP-drug constructs to specific cellular and subcellular locations via specific affinity of particular molecular targeting agents (e.g., small molecules, antibodies, peptides) for cell surface ligands are critical for the advancement of tailored NP-drug-based medicine (Scheme 1).

Our goal in this review is to provide a survey of recent advances in NMDD wherein the active targeting of NP-drug constructs to various cellular and subcellular (Scheme 1) locations (primarily to the nucleus and mitochondria of the disease-affected tissues/cells) has facilitated significant improvements in drug efficacy. We discuss this in the context of a range of NPs including liposomes, polymeric nanoparticles (polymersomes), dendrimers, and inorganic nanocomposites (semiconductor quantum dots (QDs), gold NPs (AuNPs)) for active targeting of critical diseases including cancer, rheumatoid arthritis (RA), and neurodegenerative diseases such as Alzheimer’s disease (AD). We specifically limit our discussion to those research reports published from 2015 to the present. We also highlight the current state of the use of actively targeted therapeutic NPs in the clinical setting with a brief future perspective on where we expect to see the next significant developments of actively targeted therapeutic NP-drugs.

## 2. Active Cellular Targeting of Nanoparticle (NP)-Drug Systems

During the pathological transformation from a normal cell, diseased cells adopt a number of macromolecular and morphological changes, including alterations to their shape and enzymatic profile, modifications in their surrounding microenvironment (such as pH and redox properties), and the expression of new molecules at various cellular locations [25,26,27]. The identification and functional targeting of these markers, however, is a formidable challenge [28,29]. These overexpressed markers are highly diverse and their expression patterns vary with tissue type [30], often requiring a highly specific or personalized approach for active targeting. More critically, such active targeting mechanisms based on ‘overexpression’ of specific markers are relative to the healthy cells. Other non-targeted healthy cells in certain tissues/organs of the body may also contain these specific markers, potentially even to a greater extent or in a larger volume than, for example, a tumor, leaving those non-targeted cells more vulnerable to drug toxicity [3]. However, since the implementation of NMDD, the concept of active cellular targeting based on these morphological changes and overexpressed markers has been extensively pursued and many successes have been realized [24,31]. The majority of these active cellular targeting approaches primarily function via ligand-receptor, enzyme-substrate, or antibody-antigen mediated interactions [24,30,31].

One of the most active areas for the implementation of NPs for drug delivery is the field of cancer research, with a particular emphasis on the identification of cell surface markers for NP targeting [30]. These markers include the α_ν_β_3_ integrin, myeloid antigen (CD13), cell adhesion glycoprotein (CD44), programmed death ligand-1 (CD274), folate receptor protein, vascular endothelial growth factor receptor (VEGFR), epidermal growth factor receptor 2 (HER2), and somatostatin receptors (SSTRs) overexpressed on the surface of the cancer cells [32]. The exploitation of these overexpressed cellular markers has facilitated active targeting NP-drug strategies in the context of a variety of NP platforms, and we discuss these examples in the subsequent sections.

Among the various NPs used for targeting cancer cells, liposomes are by far the most widely studied nanoplatform primarily because of their high biocompatibility, ease of surface modification, and amenability to loading both hydrophobic and hydrophilic drugs [33,34]. Due to their ability to incorporate multiple different functional phospholipids during synthesis, an array of liposomal constructs decorated with various targeting moieties have been reported for enhanced cellular targeting and uptake for improved therapeutic efficacy in cancer cells (Table A1). Specific examples here include the work of Patil et al., who formulated a folate acid-decorated liposome loaded with a prodrug version of mitomycin C [35]. In this preparation, folate was conjugated to poly(ethylene glycol) (PEG)-1,2-distearoyl-*sn*-glycerol-3-phosphoethanolamine (DSPE) lipid and displayed on the liposomal surface. These liposomes showed up to 9-fold greater membrane binding and uptake in KB HiFR epidermal carcinoma cells in vitro, and showed preferential prodrug delivery to J6456 lymphoma cells in vivo compared with non-targeted liposomes. Similarly, a folate-targeted liposomal formulation of nitrooxy-doxorubicin (N-DOX) was fabricated for overcoming P-glycoprotein (P-gp)-mediated efflux of DOX from multidrug-resistant (MDR) cells. In this preparation, a nitric oxide (NO)-releasing group on DOX overcomes MDR by inducing NO-mediated inhibition of P-gp [36]. Upon administration in a mouse model, superior cellular uptake of N-DOX to DOX-resistant MCF7 breast cancer cells coupled with enhanced anti-tumor efficacy was observed compared with Caelyx^®^ (a liposomal formulation of DOX (that is comparable to DOXIL^®^) not engineered for the defeat of MDR). Other liposomal drug formulations have incorporated multiple disparate features such as dual-targeting, enhanced cellular penetration/internalization, imaging, and immune therapy [37,38,39,40,41]. For instance, paclitaxel (PTX)-loaded hybrid liposomal NPs were decorated with anti-programmed death ligand-1 antibody to enable targeting of 4T1 breast and CT26 colon cancer cells in vitro and in vivo [37]. These liposomes were further labeled with an infrared dye (RDye800CW) and an MRI contrast agent (Gd(III)-DOTA) for dual-modality (fluorescence and MRI) imaging. Such multifunctional liposomal preparations, however, require careful incorporation of targeting ligands on their surface and often complicated chemical conjugation of lipids or PEG-lipids with the targeting moieties. Thus, recent strategies have examined the intracellular biosynthesis of natural liposome like vesicles (exosomes) that are decorated with targeting moieties [42,43,44,45,46]. For example, a biofunctionalized liposome-like nanovesicle (BLN) decorated with targeting moiety human epidermal growth factor (hEGF) was biosynthesized by Zhang et al. [43]. In this synthesis process, hEGF was genetically engineered to be immobilized on the surface of HEK 293T cells and then exosome vesicles (presenting hEGF) were induced to bud from the HEK cells using sodium deoxycholate surfactant (Figure 1A). BLN-hEGF NPs loaded with drug (indocyanine green, (ICG)) showed improved cellular uptake and photothermal efficacy in vivo in a mouse MDA-MB-468 breast cancer tumor model (Figure 1B,C) [43]. The authors also synthesized HER2-decorated BLNs that were loaded with DOX that showed better antitumor therapeutic outcomes than the clinically approved liposomal DOX NP-drug, DOXIL^®^, in HER2-overexpressing BT474 tumors in a mouse model. Kameraka et al. synthesized CD47-decorated exosomes that were engineered to carry inhibitory RNAs to target oncogenic Kras^G12D^, a common mutation in pancreatic cancer [44]. This exosomal preparation showed a superior ability to deliver RNAi and suppress tumor growth when compared to liposomes. Despite these promising results, biosynthesis of liposome-like vesicles for real clinical applications still face challenges, primarily due to the limitations of large scale synthesis and purification, reproducibility, and post-synthesis drug loading compared to traditional synthetic liposomes or other vesicles [42].

Polymeric NPs, which include self-assembled structures and dendrimers, have also been successfully exploited in NMDD for active cellular targeting for cancer (Table A1) [47,48,49,50,51,52,53,54,55]. This is primarily due to the availability of a wide variety of synthetic polymers and polymer precursors with ample chemical functionalities (such as carboxylate, amine, alkyne) for conjugations, self-assembly and physicochemical features (such a sensitivity to light, pH). NPs prepared with such polymers can be loaded with hydrophilic as well as hydrophobic drugs and conjugated to targeting moieties. Recently, DOX-loaded polytyrosine (PTN) NPs decorated with cyclized RGD peptide (cRGD-PTN-DOX) were prepared to treat colorectal cancer in vivo (Figure 2A) [48]. This preparation afforded ~3-fold better accumulation in HCT-116 tumors in mice via interaction with integrin α_ν_β_3_ compared to non-targeted NPs (Figure 2B), with over 5-fold better tolerance and improved toxicity (~6-fold lower half-maximal inhibitory concentration, IC_50_) compared with clinically used DOXIL^®^. Intravenously administrated cRGD-PTN-DOX-induced effective inhibition of HCT-116 colorectal tumor with depleting side effects (Figure 2C). The authors stated the release of the DOX at targeted cells was due to degradation of polytyrosine by proteinase K. One very desirable feature of polymeric NPs is their controllable degradation and drug release once in the intracellular environment. However, the design of such NPs requires hybrid approaches for efficient cellular internalization followed by the triggered release of drugs via intracellular (such as pH, redox, hypoxia, enzyme activities) [56,57,58,59,60,61,62,63,64,65,66] and extracellular (such as light, temperature, sound, magnetic forces) [67,68] stimulations. Liu et al. recently reported the synthesis of pH-responsive (pH 6.5) NPs consisting of poly(L-histidine) (PHIS) and an anti-tumor immune regulator resiquimod (R848) [64]. These NPs were additionally decorated with CD44-targeted hyaluronic acid (HA) and conjugated with DOX via a pH-labile hydrazone linker. Investigation with different CD44 expressing cell lines (MCF-7, 4T1) and 4T1 tumor-bearing mice suggested that deprotonation of PHIS around pH 6.5 (a pH value close to that of the tumor microenvironment) switched the nature of NPs from hydrophobic to hydrophilic. This triggered the release of R848 to exert immunoregulatory action followed by the rupture of the hydrazone linkage-bound DOX at pH ~5.5 (pH of endo/lysosomes) for accelerated release of DOX. While intracellular stimulation of drug release is versatile, the triggered release via extracellular stimulation can often offer a better degree of control over spatiotemporal drug release [13,69]. For example, Nguyen and colleagues reported methotrexate (MTX)-loaded polymeric NPs for SSTR-mediated targeted delivery [70]. These NPs were decorated with lanreotide (LT), a synthetic analog of somatostatin, on the surface of the NPs comprised of a photosensitizing polymer, polyaniline (PANI), poly(lactic-*co*-glycolic acid) (PLGA), and DSPE-PEG. This preparation showed a 2-fold greater uptake into SSTR-positive human breast adenocarcinoma MDA-MB-231 cells in BALB/c nude mice compared to non-targeted counterparts. When coupled with near-infrared (NIR), the heat-assisted burst release of MTX resulted in remarkably enhanced antitumor activity. For improved penetration and accumulation to the tumor, stimuli-responsive targeted NP constructs have been additionally functionalized with extra features such shrinkable size and vasodilation [71,72,73]. For instance, Hu and colleagues prepared DOX and ICG-conjugated dendrimer NPs decorated with enhanced tumor targeting and penetration capabilities [71]. These NPs were assembled with nitrooxyacetic acid (a NO donor as a vasodilator) and conjugated with HA for enhanced permeability. These NPs showed the ability to shrink in size from ~330 nm to ~35−60 nm via hyaluronidase degradation and had enhanced uptake when coupled with tumor internalization RGD (iRGD) peptide. This formulation showed 4.5-fold longer blood retention with 2.1-fold increased accumulation in the tumor at 36 h post-injection. Administration of this formulation in 4T1 tumor-bearing mice, followed by 808 nm laser irradiation resulted in complete ablation of the tumor via chemo- and photothermal effects.

In addition to liposomes and polymeric NPs, various inorganic nanocomposites NPs have also been investigated for active targeted cellular delivery of drugs for cancers/tumors. These include semiconductor nanocrystals or quantum dots (QDs) [74,75,76,77,78,79,80,81,82], graphene/carbon dots (GQD) [83,84,85,86,87], mesoporous silica nanoparticles (MSN) [88,89,90,91,92], gold NPs (AuNPs) [93], iron oxide nanoparticles (IONPs) [94], and lanthanide-doped upconversion NPs (UNCPs) [95]. For example, Wang et al. reported an anticancer drug, aminoflavone (AF), that was loaded on indium phosphide core/zinc sulfide shell (InP/ZnS) QDs that were decorated with anti-EGFR nanobody for targeting EGFR-overexpressing MDA-MB-468 triple-negative breast cancer (TNBC) cells [77]. Administration of these NPs for treating MDA-MB-468 TNBC bearing mouse model showed 2-fold enhanced uptake followed by more effective tumor regression compared with non-targeted counterparts. Tsai et al. synthesized UCNPs coloaded with IR-780 and the photosensitizer, 5,10,15,20-tetrakis(3-hydroxyphenyl) chlorin, and surface-modified with angiopep-2 peptide (TFFYGGSRGKRNNFKTEEYC) and cholesterol-PEG for targeting glioblastoma multiforme (GBM) [95]. In a mouse model, this construct showed low non-specific distribution with more than 1.5-fold increased accumulation in glioblastoma cells/tumors compared with a non-targeted counterpart. Externally-photoactivation for photothermal and photodynamic effects on an orthotopic glioblastoma (ALTS1C1 cells) tumor model in mouse brain resulted in an enhanced survival rate. In a final example, Cheng and colleagues prepared DOX-loaded MSNs that were coated with polydopamine (PDA) and poly(ethylene glycol)-folic acid (PEG-FA) for cervical cancer therapy. This formulation showed ~2-fold higher uptake in HeLa cells with enhanced antitumor efficacy compared to non-targeted preparation [96].

Rheumatoid arthritis (RA) is another disease that has seen the development of NP-based systems for targeted drug delivery at the cellular level [97,98,99,100]. RA is a chronic inflammatory disease with a complex pathology characterized by inflammation of the joints, destruction of the synovium, production of autoantibodies, and damage of bone and cartilage. In RA, various cellular makers such as FAR, CD44, CD64, F4/80, macrophage mannose receptor, E-selectin, intercellular adhesion molecule-1, phosphatidylserine, and matrix metalloproteinases are overexpressed on macrophages and endothelial cells of the affected tissue [101]. Zhao and colleagues formulated core-shell folate receptor (FR)-targeting and pH-responsive NPs loaded with MTX. This nanocarrier (FA-PPLNPs/MTX) was composed of PLGA–PEG–FA, pH-sensitive poly(cyclohexane-1,4-diylacetone dimethylene ketal) (PCADK), and egg lipids (Figure 3A). Targeted accumulation of the NPs followed by pH-assisted release of MTX resulted in an augmented therapeutic outcome in both in vitro and in vivo (Figure 3B) [102]. In a hybrid approach, Duan et al. delivered siRNA (NF-kB-targeted) and MTX coloaded in FA-decorated calcium phosphate/liposome to the diseased site with improved blocking of NF-kB signaling and reduced expression of pro-inflammatory cytokines in an arthritic mouse model [103]. Among the inorganic nanocomposites, there are a few demonstrations of gold [104,105,106], silver [107], and magnetic (such as superparamagnetic iron oxide) [88,108] NPs that have been used for active targeted delivery of RA drugs. Applications of these NPs in targeted delivery of anti-RA drugs are most often coupled with (1) their plasmonic nanoscale local heating and (2) ability to diagnosis the status of the disease by imaging, such as magnetic resonance imaging (MRI). For example, Kim and colleagues prepared MTX-loaded poly(lactic-*co*-glycolic acid) (PLGA) gold (Au)/iron (Fe)/Au half-shell NP for chemo-photothermal therapy and multimodal imaging of RA [104]. Overall, the development of NP-drug systems against cellular markers for RA is still in the early stages, likely due to the pathological complexity of RA with the compatible design of the NP-drug system conjugated with targeting ligands.

Like cancer and RA, Alzheimer’s disease (AD) has garnered significant attention in the development of active targeted cellular delivery of NP-drug systems. With no preventive treatment available, a major obstacle to treat AD is the delivery of drugs across the blood-brain barrier (BBB) to the central nervous system (CNS) [109]. Recent advances with various NP systems have shown promise to cross this barrier [110]. For example, Clark et al. prepared AuNPs (80 nm) coated with PEG-Tf having an acid-cleavable linkage between Tf and the NP. These AuNPs were designed to bind to Tf receptors on the blood side of the BBB. Tf-TfR interactions were abrogated when acid-induced cleavage occurred during transcytosis, allowing the release of the AuNPs into the brain. This resulted in an ~3-fold increase in the availability of these AuNPs in the brain parenchyma (mouse model) compared with AuNPs with non-cleavable linkage [111]. However, because the surface of AD^+^ neuronal cells are undifferentiated morphologically from healthy neuronal cells [112], the lack of suitable cell markers has made targeting NPs in AD a considerable challenge. Currently, the clinically accepted pathological marker of AD is the accumulation of intracellular neurofibrillary tangles (NFT) and extracellular amyloid-beta (Aβ) plaques that exert neurotoxicity and are believed to be responsible for AD [113,114]. Thus, NP-based diagnostic and therapeutic strategies that target Aβ-induced neurotoxicity are an attractive approach to treat AD. Ahlschwede et al. prepared curcumin-loaded PLGA NPs coated with cationic peptide (K16ApoE) that is capable of crossing BBB and binding to amyloid plaques [115]. This peptide is composed of 16 lysine residues and amino acids 151–170 of the low-density lipoprotein receptor (LDLR)-binding segment for targeting of vasculotropic DutchAβ40 peptide accumulated in the cerebral vasculature. This preparation showed up to 10-fold higher accumulation in various regions of DutchAβ40 treated mice as compared to the NPs without K16ApoE decoration. Zheng and colleagues prepared a H102 (a β-sheet breaker peptide, HKQLPFFEED)-loaded PEG-poly(lactic acid) (PEG-PLA) NP that was surface-modified with the peptides TGN (TGNYKALHPHNGC) and QSH (QSHYRHISPAQVC) for BBB penetration and Aβ42 ligand binding, respectively [116]. This dual-targeted NP (TQNP) was able to cross the BBB with 4-fold more accumulation than control NP. TQNP also showed better Aβ plague targeting efficacy in vitro (bEnd.3 cells) and in vivo (5XFAD mice). The authors showed better therapeutic efficacy of this preparation via a decrease in amyloid plaques, a reduction in tau protein phosphorylation and improvement in spatial learning and memory of transgenic mice than NPs modified with a single ligand [116]. AuNP-induced heating has also been used to target Aβ aggregation in neuronal cells in vitro and in vivo [117,118]. Liu et al. showed NIR-induced heating of gold nanorods (GNRs) decorated with anti-Aβ a single chain variable fragment (*scFv* 12B4) and the thermophilic amyloid-degrading enzyme, acylpeptide hydrolase (APH) [118]. This formulation showed both rapid detection of Aβ aggregates and mediated the disassembly of Aβ aggregates and inhibited Aβ-mediated toxicity in *Caenorhabditis elegans*.

## 3. Active Subcellular Targeting of NP-Drug Systems

While the NP-drug approaches highlighted thus far show utility in cell killing simply by delivering the drug cargo to the cell interior, other drug formulations require the release of the drug directly at the subcellular site/organelle of action (e.g., nucleus, mitochondria, lysosome, endoplasmic reticulum, Golgi body) (Scheme 1). Although cell surface markers have proven effective for the active targeting to and improvement of drug efficacy in desired cell types (vide supra), this is not always sufficient to drive the delivery of drugs to specific subcellular locations/organelles [119,120]. This has triggered increasing efforts that are focused on developing more sophisticated NP-drug designs to target the internalized NP-drug complexes to the intracellular site of action. A general design theme of such subcellular targeted NP-drug systems is that they possess dual targeting (cell-type specificity and subcellular localization) capability coupled with multistage, triggered the release of the on-board drug cargo [38,39]. In this section, we highlight recent examples (Table A2) of NP-drug systems that have achieved augmented therapeutic outcomes as a result of targeted delivery to subcellular organelles such as the nucleus and mitochondria of diseased cells and tissues.

### 3.1. Targeting of NP-Drug Systems to the Nucleus

The nucleus is recognized as an ideal center for targeted drug delivery as it is the central repository of and the initial distribution center for the cell’s genetic information [120,121]. The synthesis of functional cellular proteins begins with the transcription of encoded genes from DNA into messenger RNA (mRNA) in the nucleus. These mRNAs are then translocated to the cytosol where the translation machinery assembles the proteins according to the mRNA “blueprint”. Passive, non-targeted delivery of drugs to the nucleus requires that they enter the nucleus via the nuclear pore (~10 nm diameter) and/or the compromised nuclear membrane during mitosis. Nuclear-targeted NP-drug constructs that can enter through the nuclear pore are potentially more efficient, but this process is limited by the channel size (~10 nm) of the nuclear pore complex (NPC) [121]. In the cell, materials destined for active nuclear transport (e.g., transcription factor proteins) contain a nuclear localization signal (NLS) that simultaneously binds to the protein cargo and to the active transport proteins (importins) that shuttle the cargo to the to NPC, which actively translocates cargos (even those larger than 10 nm) into the nucleus. The canonical NLS is a short, positively-charged peptide sequence rich in basic amino acids (e.g., the transactivator of transcription (TAT) peptide (YGRKKRRQRRR)) [122]. Recent studies have shown that NP-drug constructs bearing such peptide motifs can translocate NPs and appended drug cargos into the nucleus of cells and release the drug inside the nucleus which can dramatically (1) improve the efficacy of therapy, (2) circumvent multidrug resistance, and (3) reduce off-target toxicity [123,124,125,126].

For cancer, nuclear-targeted delivery is ideal for many chemotherapeutics drugs including DOX, primarily because they exert their function by reacting directly with the DNA or by inhibiting DNA topoisomerase to induce cell death [123,127,128,129]. However, drug delivery to the nucleus is challenging because of the evolution of MDR whereby cancer cells express membrane-resident P-glycoprotein (P-gp) pumps that actively efflux various anticancer drugs from the cytosol to the extracellular environment [130,131,132]. Recently, Han et al. reported DOX-loaded pH-sensitive core-shell silica NPs (CSNPs) that were decorated with galactose (Gal) and TAT for targeting hepato-carcinoma ascites cells (H-22) and the nucleus via asialoglycoprotein receptor (ASGPR) and NLS mediated interactions, respectively (Figure 4A) [133]. These CSNPs were further functionalized with an acid-cleavable PEG linker for enhanced in vivo circulation and reduced phagocytic clearance. The authors described CSNPs as a triple-stage targeted delivery approach in where the NPs undergo PEG detachment via acidic hydrolysis in tumor microenvironment (pH 6.5), then the exposed Gal ligands endow CSNPs with active internalization into hepato-carcinoma cells. Upon internalization, endosomes and lysosomes (pH ~5.0) triggered conversion of the anionic shell into positive charges, leading to core-shell disassembly and subsequent TAT-mediated translocation to the nucleus where DOX is delivered (Figure 4A). When administered by i.v. to H-22 tumor-bearing mice, CSNPs showed up to 40-fold higher tumor accumulation than free DOX, resulting in 91.1% tumor inhibition ratio (TIR) compared to that of 43.8% for free DOX (Figure 4B). Similar TAT functionalized iron oxide NPs (IONPs) were reported for nuclear-targeted delivery for photothermal therapy (PTT) [134]. These IONPs were conjugated with transferrin for active cellular targeting of lung cancer cells (A549 cells). These targeted IONPs showed 40-fold higher accumulation to the nuclei of the A549 cells in vitro and better PTT outcomes of A549 tumor-bearing nude mice compared to non-targeted particles [134]. In a more sophisticated, size-shrinkable approach, Fan et al. reported the use of a poly-l-lysine peptide as an NLS for nuclear delivery of an iridium(III) metallodrug. The NPs (~150 nm), which were decorated with FA and a degradable PEG-shell for active and enhanced nuclear targeting, were designed to stay protected in the PEG shell under physiological conditions (pH 7.4) [135]. Once internalized into intracellular acidic endo/lysosomes (pH < 5.5), pH-responsive linkages in the PEG chains are cleaved, yielding smaller NPs (~40 nm) with the oligo-l-lysine exposed on the NP surface. The smaller NPs, with the exposed NLS, then translocate into the nucleus via the nucleopore, resulting in a 20-fold lower IC_50_ to HeLa cells in vitro and better therapeutic outcomes with 3.7-fold better TIR in vivo compared to the free drug. In a similar multistage approach, core-shell iron oxide-based NPs and mesoporous silica NPs were loaded with tirapazamine (TPZ), a hypoxia-activated prodrug that causes apoptosis and surface decorated with anti-CD133 antibody and TAT peptide for targeting cancer cells and the nucleus, respectively [136]. Unlike the aforementioned example, upon internalization, the TAT peptide was exposed by the cleavage of thermo-sensitive azo linkages via the application of an external alternating magnetic field, resulting in enhanced nuclear accumulation and better therapeutic efficacy compared to non-target counterparts.

### 3.2. Targeting of NP-Drug Systems to the Mitochondria

The mitochondrion plays a number of critical roles in cellular physiology and homeostasis [119,120]. These include oxidative phosphorylation to provide energy to the cell in the form of ATP, the regulation of reactive oxygen species (ROS), and the control of intracellular calcium ion concentration. Mitochondrial malfunctions can result in the onset of a number of diseases including cancer, Alzheimer’s, Parkinson’s, and diabetes. Thus, the mitochondrion has emerged as a viable target for drugs for enhanced therapeutic action aimed at 1) apoptosis-mediated cell killing either by delivery of the chemotherapeutic drug to mitochondrial DNA or increasing oxidative stress and 2) protection of the cell by scavenging ROS in the tissue of interest [137,138].

Despite many clinically approved drugs that act on mitochondria, there are only a few examples of drugs that can actively partition specifically into the mitochondria. Drug access into the mitochondrion is challenging owing to its complex membrane structure coupled with a highly negative membrane potential (−150 to −180 mV) that prohibits the entry of small molecules (particularly anionic) to the inner space [120]. One strategy that has documented success is the use of cationic, lipophilic peptides/small molecules that facilitate accumulation to the anionic mitochondrial membrane via electrostatic interaction, followed by translocation through the lipophilic membrane to the mitochondrial matrix [139,140,141,142,143]. Several mitochondrial-targeting sequence (MTS) motifs have been identified [144,145,146]. For example, Jian et al. showed a pH-dependent charge reversal approach with the MTS d[KLAKLAK]_2_ (KLA) that was conjugated to 3-dimethylmaleic anhydride (DMA) (KLA-DMA). This KLA-DMA further conjugated with DSPE lipid (DSPE-KLA-DMA (DKD)) to make a liposomal formulation of PTX (DKD/PTX-Lip) for mitochondrial targeting of taxol-resistant lung cancer cells (A549/Taxol) [144]. At extracellular pH (~6.8), the KLA-DMA peptide on the liposomal surface converted to neutral from negative (pH = 7.4) by cleaving amide linkage between KLA and DMA, facilitating cellular internalization (Figure 5A). After cellular uptake, at pH ~5.5 KLA became positive, and the positively charged liposomes were accumulated in mitochondria where PTX was released. This resulted in 4-fold higher mitochondrial accumulation and 5.5-fold increased toxicity to A549/Taxol cells compared to traditional liposomal formulation of PTX. When administered (i.v.) to A549/Taxol tumor-bearing mice this formulation showed TIR 86.7% compared to that of 48.7% when treated with non-targeted liposomes (Figure 5B,C). MTS-mediated active mitochondrial targeting also has been reported for combinatorial delivery of photothermal/chemotherapeutic agents. Recently, Chen et al. co-encapsulated gold nanostar (AuNS) and DOX in HA protective shell for tumor-targeting synergistic photothermal/chemo-therapy [147]. In this formulation, two peptides (cationic octaarginine (R_8_) and mitochondria-targeting pro-apoptotic KKKLAKLAKKLAKLAK-C conjugated with triphenylphosphonium) were co-decorated on AuNS via Au-S bond. This preparation showed enhanced CD44-mediated recognition and uptake to squamous cell carcinoma (SCC-7) and DOX-resistant human breast cancer cells (MCF-7/DOX). Followed by digestion with hyaluronidase (HAase), this formulation releases DOX in the cytosol and navigates to the mitochondrial where the photothermal effect was induced via NIR irradiation. Synergistic photothermal/chemo-therapy resulted in enhanced inhibition of non-resistant or resistant tumor cells both in vitro and in vivo [147]. Chan et al. prepared DOX-loaded nanodiamonds (ND) that were decorated with FA and dual (cell-permeable and MTS) functional peptide MLSLRQSIRFFKPATRTLCSSRYLL for targeting the mitochondria of DOX resistant human breast cancer cells MCF-7 [148]. Clearly, MTS-mediated mitochondrial targeting has become a feasible and biocompatible method, but it involves the expensive synthesis of peptide sequence followed by decoration of NP using chemical conjugation reactions during which most often it requires controlling their aggregation-induced precipitation via electrostatic interaction.

Still, other mitochondrial targeting approaches have utilized small molecule ligands. A common theme that has emerged is the use of the triphenylphosphonium (TPP) moiety [149,150]. Song et al. reported the use of TPP for the mitochondrial delivery of mesoporous Ce6-loaded Au@Pt NPs for combined photodynamic therapy (PDT)/PTT. The Au@Pt NPs were demonstrated to augment the catalytic conversion of H_2_O_2_ into O_2_ for improved PDT efficacy [151]. Similarly, Yang et al. reported on silica nanoreactors encapsulated with catalase, a water-soluble H_2_O_2_-degrading enzyme, for the enhancement of Ce6-induced PDT through the elevation of oxygen levels in the mitochondria of tumor cells [152]. More recently, the natural product ligand glycyrrhetinic acid (GA), a pentacyclic triterpenoid obtained from Glycyrrhiza glabra, has been reported as a mitochondrial targeting agent [153]. Zhang et al. showed that GA-functionalized graphene oxide NPs loaded with DOX efficiently accumulated in the mitochondria of cancer cells with greatly improved therapeutic efficacy in vitro (HepG2 cells) and in vivo (HepG2 tumor-bearing mice) with decreased off-target toxicity.

## 4. Conclusions and Future Perspective

From the survey of the recent studies presented above, it is clear that cellular and subcellular targeted delivery of NP-drug complexes is an ever-expanding area of drug development. The use of NPs for targeted cellular delivery not only affords the improved therapeutic index of drugs with lower drug dosages, but it can also circumvent side effects and the avoidance of MDR. It is also clear that the level of complexity and sophistication of these NP-based materials is also growing exponentially. Still, many hurdles remain to be overcome before both cellular and subcellular targeted NP-drug systems reach their full clinical potential. These primarily include the synthesis of targeted NPs with precise targeting and a clear understanding of targeting mechanisms, acceptable reproducibility, biodegradability, clearance of the NPs, and minimal off-target toxicity. Cumulatively, these are extremely challenging tasks that will require a focused approach for each respective NP-drug system aimed at a specific disease.

Currently, where do the active targeted NP-drug constructs stand in the clinical trial setting? Most of the currently approved NP-drugs for clinical trials are simple formulations of liposomes or polymeric NPs that are delivered by passive targeting [154]. There are few active cellular-targeted NP-drugs constructs that have entered the clinical trial (e.g., Trial# NCT03774680, NCT02979392) pipeline on ClinicalTrials.gov for treating/diagnosing cancer. For example, cetuximab loaded ethylcellulose polymer (Trial# NCT03774680) that was decorated with somatostatin analog for targeting colon cancer is in Phase 1. However, as of now, none of the subcellular organelle-targeting NP-drugs have reached clinical trials yet. Clearly, more understanding and control over preparations and navigation of targeted NP-drug constructs to the organelles of the disease-affected tissue is highly desired. Overall, such NP-drug constructs are still in infancy, and to reach the clinical trial, many basic science questions will have to be answered even after successful demonstration of their significant advantages over non-targeted preparation.

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
