# Peer review of "Active Cellular and Subcellular Targeting of Nanoparticles for Drug Delivery"

_pharmaceutics, 2019, doi:10.3390/pharmaceutics11100543_

Round 1
Reviewer 1 Report
In the present MS, the “Active Cellular and Subcellular Targeting of Nanoparticles for Drug Delivery” was reviewed.
Unfortunately, this MS was disordered.
The title was casual, “1. Introduction; 2. Active Cellular Targeting of NP-drug Conjugates; 2. Active Subcellular Targeting of NP-drug Systems; 5. Conclusion and Future Perspective” What is the difference between “NP-drug Conjugates” and “NP-drug Systems”? In the “Active Subcellular Targeting of NP-drug Systems” section, author divided 3 subsection, “3.1. Targeting of NP-Drugs to the Nucleus; 3.2. Targeting of NP-Drugs to Mitochondria”, however, there was no subsection in “Active Cellular Targeting of NP-drug Conjugates”. The targeting mechanism should be introduced and classified in each section and subsection.
Author Response
Reviewer 1
1. In the present MS, the “Active Cellular and Subcellular Targeting of Nanoparticles for Drug Delivery” was reviewed.
Unfortunately, this MS was disordered.
The title was casual, “1. Introduction; 2. Active Cellular Targeting of NP-drug Conjugates; 2. Active Subcellular Targeting of NP-drug Systems; 5. Conclusion and Future Perspective” What is the difference between “NP-drug Conjugates” and “NP-drug Systems”? In the “Active Subcellular Targeting of NP-drug Systems” section, author divided 3 subsection, “3.1. Targeting of NP-Drugs to the Nucleus; 3.2. Targeting of NP-Drugs to Mitochondria”, however, there was no subsection in “Active Cellular Targeting of NP-drug Conjugates”.
Author response: We thank the Reviewer for bringing to our attention to these important ordering issues. We have addressed these in the revised manuscript.
First, we replaced, in Section 2 (p. 3), we have replaced ‘Active Cellular Targeting of NP-Drug Conjugates’ with ‘Active Cellular Targeting of NP-Drug Systems’ which encompasses NP-based drug (NP-drug) formulations. The title of this section is now consistent with the title for Section 3 (Active Subcellular Targeting).
Second, the revised manuscript has been re-organized with the following sections (in order): 1. Introduction; 2. Active Cellular Targeting of NP-Drug Systems; 3. Active Subcellular Targeting of NP-Drug Systems; 3.1. Targeting of NP-Drug Systems to the Nucleus; 3.2. Targeting of NP-Drug Systems to the Mitochondria; and 4. Conclusion and Future Perspective.
2. The targeting mechanism should be introduced and classified in each section and subsection.
Author response: We again thank the Reviewer for this insightful comment. In the original manuscript, the authors believed each targeting section/subsection started with adequate discussion relevant to targeting strategies and mechanisms that followed with specific examples of these targeting strategies for various NP-drug formulations. To more generally introduce the reader to the various targeting mechanisms, in Section 2, Active Cellular Targeting of NP-Drug Systems (p.3), we have added the following line: “The majority of these active cellular targeting approaches primarily function via ligand-receptor, enzyme-substrate, or antibody-antigen mediated interactions [24,30,31]”.
Reviewer 2 Report
This review briefly describes current progress on active cellular and subcellular targeting nanoparticle-based drug delivery systems with examples in literature published within 5 years. The review is well written and well summarized current issues and challenges. Thus, the review would be interesting for the researchers to study drug delivery. However, minor concerns should be solved prior to its publication.
Doxil and Caelyx have been used in this review. However, although two brand names are different because their selling regions are different, they have the same formulation. Thus, the authors should describe that the formulation is same to avoid misunderstanding of some researchers. (4 page) The authors described stimuli and categorized them as intercellular and extracellular stimulation. However, the terms seem to be wrong. In general, the former and the latter have been used as “internal” and “external” triggers, respectively. The reviewer strongly suggest that the terms used by the authors should be replaced. (5 page) As an example for RA, the authors described NF-kB targeted drug delivery systems. However, NF-kB is a transcriptional factor in cytoplasm and nucleus. Thus, the authors should clearly indicate whether the example is for cellular targeting or subcellular targeting. (6 page) As an example for AD, the authors described BBB-penetrating peptide decorate NPs (Ref 114). However, it is not an example for targeting AD-positive neuronal cells. Thus, the authors should explain appropriately the example.
Author Response
This review briefly describes current progress on active cellular and subcellular targeting nanoparticle-based drug delivery systems with examples in literature published within 5 years. The review is well written and well summarized current issues and challenges. Thus, the review would be interesting for the researchers to study drug delivery. However, minor concerns should be solved prior to its publication.Author response: We thank the Reviewer for the compliments.
Doxil and Caelyx have been used in this review. However, although two brand names are different because their selling regions are different, they have the same formulation. Thus, the authors should describe that the formulation is same to avoid misunderstanding of some researchers.
Author response: Clarification has been made in the revised manuscript on p. 3 by the modification of the sentence: “…compared with Caelyx® (a liposomal formulation of DOX (that is comparable to DOXIL®) not engineered for defeat of MDR)”.
(4 page) The authors described stimuli and categorized them as intercellular and extracellular stimulation. However, the terms seem to be wrong. In general, the former and the latter have been used as “internal” and “external” triggers, respectively. The reviewer strongly suggest that the terms used by the authors should be replaced.Author response: Thank you for finding this typographical error. On p. 4, the term “intercellular” has been replaced with “intracellular”. The authors, in a previously published manuscript (Ther. Deliv. 2016, 7, 335-352), have discussed triggered drug release via various intracellular and extracellular stimulation, and this reference is cited In the current manuscript (p. 4, line 190) to clarify the difference between these two types of triggered drug release. No other changes have been made to the manuscript.
(5 page) As an example for RA, the authors described NF-kB targeted drug delivery systems. However, NF-kB is a transcriptional factor in cytoplasm and nucleus. Thus, the authors should clearly indicate whether the example is for cellular targeting or subcellular targeting.Author response: In the revised manuscript, NF-kB has been removed from the list of the cellular marker for targeting RA. All RA related NP-drug systems (for example, ref 102, 103) described in this section are targeted to the cell surface via receptors such as folic acid receptor (FAR). Thus, this section belongs to the “active cellular targeting”.
(6 page) As an example for AD, the authors described BBB-penetrating peptide decorate NPs (Ref 114). However, it is not an example for targeting AD-positive neuronal cells. Thus, the authors should explain appropriately the example.Author response: On p.6, the specific example described is PLGA NPs coated with a cationic peptide (K16ApoE) that is capable of crossing BBB and binding to amyloid plaques. For clarification we have modified the relevant section in the revised manuscript (p. 6) as follows:
‘Ahlschwede et al. prepared curcumin-loaded PLGA NPs coated with cationic peptide (K16ApoE) that is capable of crossing BBB and binding to amyloid plaques [115].’
Reviewer 3 Report
In this paper, Okhil et al. summarized active targeting strategies for nanoparticles focused on cellular and subcellular targeting. The contents are good and valuable considering recent growing interests in subcellular targeting of nanoparticles. Overall figures and contents are well arranged, and I recommend the publication after minor revision as below.
Side effect of targeting with biological ligand is also an important problem. For example, HA receptors are overexpressed in liver, and RGD and folate receptors are also abundant in certain organs. I recommend explaining and discussing more about it after finding references.
The authors summarized already-performed researches very well. However, discussion or future perspectives are not sufficient. What are the main hurdles for cellular and subcellular targeting and how can we overcome them?
The authors mentioned about MDR in Abstract, but there are too little contents related with MDR. It would be helpful to reinforce them with more reports and references.
In AD therapy, crossing BBB is highly important before cellular or subcellular targeting. I recommend discussing more about that with reference below.
Clark, A.J.; Davis, M.E. Increased brain uptake of targeted nanoparticles by adding an acid-cleavable linkage between transferrin and the nanoparticle core. Proc. Natl. Acad. Sci. USA 2015, 112, 12486–12491.
Author Response
In this paper, Okhil et al. summarized active targeting strategies for nanoparticles focused on cellular and subcellular targeting. The contents are good and valuable considering recent growing interests in subcellular targeting of nanoparticles. Overall figures and contents are well arranged, and I recommend the publication after minor revision as below.Author response: We would like to thank the Reviewer for the compliments.
Side effect of targeting with biological ligand is also an important problem. For example, HA receptors are overexpressed in liver, and RGD and folate receptors are also abundant in certain organs. I recommend explaining and discussing more about it after finding references.
Author response: We thank the Reviewer for this insightful comment. Based on the published literature, we have added the following on p. 3 in the revised manuscript:
More critically, such active targeting mechanisms based on ‘overexpression’ of specific markers are relative to the healthy cells. Other non-targeted healthy cells in certain tissues/organs of the body may also contain these specific markers, potentially even to a greater extent or in a larger volume than, for example, a tumor, leaving those non-targeted cells more vulnerable to drug toxicity [3]. However, since the implementation of NMDD, the concept of active cellular targeting based on these morphological changes and overexpressed markers has been extensively pursued and many successes have been realized [24,31].
The authors summarized already-performed researches very well. However, discussion or future perspectives are not sufficient. What are the main hurdles for cellular and subcellular targeting and how can we overcome them?
Author response: The Reviewer is correct in this assessment. We have included the following in the Conclusion and Future Perspective in the revised manuscript:
‘Still, many hurdles remain to be overcome before both cellular and subcellular targeted NP-drug systems reach their full clinical potential. These primarily include the synthesis of targeted NPs with precise targeting and a clear understanding of targeting mechanisms, acceptable reproducibility, biodegradability, clearance of the NPs, and minimal off-target toxicity. Cumulatively, these are extremely challenging tasks that will require a focused approach for each respective NP-drug system aimed at a specific disease.’
The authors mentioned about MDR in Abstract, but there are too little contents related with MDR. It would be helpful to reinforce them with more reports and references.
Author response: The Reviewer is correct on this point. Accordingly, we have listed in the Conclusion and Future Perspective overcoming MDR as one of the benefits offered by nanoparticle (NP)-mediated drug delivery. Additionally, throughout the manuscript, we discuss MDR in the context of relevant examples of NP-drug systems in multiple sections, including one figure (Figure 5). Examples include: 1) p. 3, line 130, ref 36; 2) p. 8, line 376, ref 143 (Figure 5); 3) p. 8 line 391 ref 146; 4) in Table 1, ref 61, ref 157; and 5) in Table 2 ref 142, ref 147, ref 163. Given the main focus of this manuscript is to highlight targeted cellular and subcellular delivery of NP-drug systems, the authors believe the current version of the manuscript contains adequate reports relevant to MDR.
In AD therapy, crossing BBB is highly important before cellular or subcellular targeting. I recommend discussing more about that with reference below.
Clark, A.J.; Davis, M.E. Increased brain uptake of targeted nanoparticles by adding an acid-cleavable linkage between transferrin and the nanoparticle core. Proc. Natl. Acad. Sci. USA 2015, 112, 12486–12491.
Author response: We appreciate the Reviewer for pointing us to this excellent article. We have included it in the discussion in the relevant section as follows:
‘For example, Clark et al. prepared AuNPs (80 nm) coated with PEG-Tf having an acid-cleavable linkage between Tf and the NP. These AuNPs were designed to bind to Tf receptors on the blood side of the BBB. Tf–TfR interactions were abrogated when acid-induced cleavage occurred during transcytosis, allowing the release of the AuNPs into the brain. This resulted in an ~3-fold increase in the availability of these AuNPs in the brain parenchyma (mouse model) compared with AuNPs with non-cleavable linkage [111].’
Reference 111 has been added to the revised bibliography.
Reviewer 4 Report
The manuscript entitled “Active Cellular and Subcellular Targeting of Nanoparticles for Drug Delivery” submitted by Okhil K. Nag and James B. Delehanty reviews recent advances in active targeting of nanoparticles for drug delivery. This review provides an interesting survey of the nanoparticle mediated drug delivery and highlights the therapeutic use in diseases such as cancer, rheumatoid arthritis and Alzheimer. This review is well constructed and gives a large overview of the actual knowledge of active targeting. I recommend its publication in Pharmaceutics journal after minor corrections cited below.
1- In the abstract, the authors claim to review the advancements of the field over the last five years. However, in their references, we can find articles older than 5 years. for example, ref 5 (2007), ref 10 (2005) etc... This sentence must be changed.
2- Page 4, line 148, authors must delete Figure 1D which is not present in the figure 1.
3- In tables 1 and 2, the abbreviations must be written in alphabetical order for easy reading.
4- Some abbreviations must be explained such as DKD, PTX in page 8 and PTD in page 9.
Author Response
The manuscript entitled “Active Cellular and Subcellular Targeting of Nanoparticles for Drug Delivery” submitted by Okhil K. Nag and James B. Delehanty reviews recent advances in active targeting of nanoparticles for drug delivery. This review provides an interesting survey of the nanoparticle mediated drug delivery and highlights the therapeutic use in diseases such as cancer, rheumatoid arthritis and Alzheimer. This review is well constructed and gives a large overview of the actual knowledge of active targeting. I recommend its publication in Pharmaceutics journal after minor corrections cited below.
Author response: We thank the Reviewer for the careful examination of the manuscript and for the positive feedback.
1- In the abstract, the authors claim to review the advancements of the field over the last five years. However, in their references, we can find articles older than 5 years. for example, ref 5 (2007), ref 10 (2005) etc... This sentence must be changed.
Author response: We appreciate this comment. However, these references were cited in the manuscript for supporting the background information relevant to nanoparticle-mediated drug delivery and not presented as featured, highlighted recent examples of NP-drug systems. No change has been made to the manuscript.
2- Page 4, line 148, authors must delete Figure 1D which is not present in the figure 1.
Author response: Figure 1D has been removed from the revised manuscript.
3- In tables 1 and 2, the abbreviations must be written in alphabetical order for easy reading.
Author response: Abbreviations for Table 1 and Table 1 have been ordered alphabetically in the revised manuscript.
4- Some abbreviations must be explained such as DKD, PTX in page 8 and PTD in page 9.
Author response: On p. 8, DKD has been defined in the revised manuscript. PTX was defined in the original manuscript on p. 3.
PTD has been changed to PDT; it was defined in the original manuscript page 9.
Round 2
Reviewer 1 Report
Accept